# Comparative Analysis of Diagnostic Performance of Automatic Breast Ultrasound, Full-Field Digital Mammography and Contrast-Enhanced Mammography in Relation to Breast Composition

**DOI:** 10.3390/biomedicines11123226

**Published:** 2023-12-06

**Authors:** Marta Ewa Pawlak, Wojciech Rudnicki, Anna Borkowska, Karolina Skubisz, Rafał Rydzyk, Elżbieta Łuczyńska

**Affiliations:** 1Diagnostic Imaging Department, University Hospital in Cracow, 30-688 Cracow, Poland; pawlaczkamarta@gmail.com; 2Department of Electroradiology, Jagiellonian University Medical College, 30-688 Cracow, Poland; wrudnicki@cmuj.pl (W.R.); a.borkowska@uj.edu.pl (A.B.); karolina.skubisz@uj.edu.pl (K.S.); 3Diagnostic Imaging Department, 5th Military Clinical Hospital in Krakow, 30-901 Cracow, Poland

**Keywords:** mammography, contrast-enhanced mammography, automated breast ultrasound, breast cancer

## Abstract

This single center study includes a comparative analysis of the diagnostic performance of full-field digital mammography (FFDM), contrast-enhanced mammography (CEM) and automatic breast ultrasound (ABUS) in the group of patients with breast American College of Radiology (ACR) categories C and D as well as A and B with FFDM. The study involved 297 patients who underwent ABUS and FFDM. Breast types C and D were determined in 40% of patients with FFDM and low- energy CEM. CEM was performed on 76 patients. Focal lesions were found in 131 patients, of which 115 were histopathologically verified. The number of lesions detected in patients with multiple lesions were 40 from 48 with ABUS, 13 with FFDM and 21 with CEM. Compliance in determining the number of foci was 82% for FFDM and 91% for both CEM and ABUS. In breast types C and D, 72% of all lesions were found with ABUS, 56% with CEM and 29% with FFDM (*p* = 0.008, *p* = 0.000); all invasive cancers were diagnosed with ABUS, 83% with CEM and 59% with FFDM (*p* = 0.000, *p* = 0.023); 100% DCIS were diagnosed with ABUS, 93% with CEM and 59% with FFDM. The size of lesions from histopathology in breast ACR categories A and B was 14–26 mm, while in breast categories C and D was 11–37 mm. In breast categories C and D, sensitivity of ABUS, FFDM and CEM was, respectively, 78.05, 85.37, 92.68; specificity: 40, 13.33, 8.33; PPV (positive predictive value): 78.05, 72.92, 77.55; NPV (negative predictive value): 40, 25, 25, accuracy: 67.86, 66.07, 73.58. In breast categories A and B, sensitivity of ABUS, FFDM and CEM was, respectively, 81.25, 93.75, 93.48; specificity: 18.18, 18.18, 16.67; PPV: 81.25, 83.33, 89.58; NPV: 18.18, 40, 25; accuracy: 69.49, 79.66, 84.62. The sensitivity of the combination of FFDM and ABUS was 100 for all types of breast categories; the accuracy was 75 in breast types C and D and 81.36 in breast types A and B. The study confirms the predominance of C and D breast anatomy types and the low diagnostic performance of FFDM within that group and indicates ABUS and CEM as potential additive methods in breast cancer diagnostics.

## 1. Background

Breast cancer is the most frequently occurring malignant cancer among women, being the reason for 685,000 deaths of women in the world [1]. Breast cancer diagnosis and treatment pose a huge medical and social challenge, with treatment success depending mostly on early detection. According to the data derived from the Validation Study of the American Joint Committee on Cancer Eighth Edition Prognostic Stage Compared With the Anatomic Stage in Breast Cancer, the 5-year survival rate for patients with breast cancer stage I is 98–100%, while for patients with stage III cancer is only 66–98%, which emphasizes the role of early diagnosis in the cancer treatment process [2]. 

Traditional 2D mammography (full-field digital mammography—FFDM) is the only diagnostic method with a proven impact on the reduction in breast cancer mortality rate [3,4,5]. The introduction of screening with FFDM caused death rates to drop by approximately 44% [6].

The ACR, with its BI-RADS (Breast Imaging Reporting and Data System), introduced a commonly used mammographic density (MD) classification dividing breasts into subcategories depending on fatty and fibroglandular tissue percentage. There are four descriptors: type A: the breasts are almost entirely fatty (<=25% of fibroglandular tissue), type B: there are scattered areas of fibroglandular density (26–50%), type C: the breasts are heterogeneously dense, which may obscure small masses (51–75%), type D: the breasts are extremely dense, which lowers the sensitivity of mammography (76–100%) [7]. Figure 1 presents the breast composition types with mammography and Figure 2 presents the breast composition types with ABUS.

About 43% of women aged 40–74 have breasts of glandular type. It mostly pertains to women younger than 40, as glandular tissue is replaced with fatty tissue over the course of time. Moreover, that type of breast anatomy is related to genetic predisposition, low BMI and application of contraceptive treatment or hormone replacement therapy. Approximately 59% of women with normal BMI have dense breasts [8]. The above data indicate the prevalence of this type of breast composition. Fibroglandular dense breast type poses an independent risk factor for breast cancer (the risk is about 4–6 times greater in comparison with other breast types), and accounts for a higher mortality rate. Within the group of women with dense breasts, interval breast cancers are diagnosed more often, as well as locally and generally advanced cancers, which may require more aggressive treatment with a low probability of a positive outcome [9,10].

Studies conducted so far have shown the efficiency of screening FFDM performance as not satisfactory within certain groups of patients and thus requiring complementary screening with another method. Referring to the focal lesions detection, the most problematic are the patients with breast categories C and D, where FFDM sensitivity is alarmingly low, and patients with a genetic predisposition to breast cancer [11,12,13,14,15,16]. These factors are the reasons for conducting research focused on the described group of patients, seeking an adequate diagnostic method which may support or replace FFDM in screening.

Within the last few years, multiple studies whose aim was to improve focal lesions diagnostics in patients with dense breasts have been carried out. The analyses pertained mainly to handheld ultrasonography (HHUS), digital breast tomosynthesis (DBT) and breast magnetic resonance imaging (BMRI). The added value of DBT is based on the possibility of acquiring a series of thin-slice images due to the X-ray tube movement around a compressed breast. Thin-slice images greatly reduce the summation phenomenon characteristic for FFDM and limit its efficiency in dense breast diagnostics. The latest ECIBC recommendations refer to the application of DBT in screening for women whose breasts were described as dense on the previously performed FFDM [17]. 

Another method which is a developed mammography technique is CEM—contrast- enhanced mammography. Basically, it is a mammography performed according to a standard protocol reinforced with iodine contrast medium administration and low- and high-energy image acquisition. Low-energy images are equivalent to the images obtained with FFDM, whereas high-energy images show areas of contrast enhancement. These two types of images are both submitted to digital processing by the subtraction of low-energy images from high-energy images, which results in creating images showing exclusively contrast-enhancement areas, whereas glandular tissue is subject to attenuation. CEM has higher sensitivity and accuracy than FFDM, particularly in patients with dense breast composition and patients with a medium risk of breast cancer incidence. Moreover, CEM is more precise in estimating the real size of the lesions than both FFDM and HHUS [18,19,20,21]. Using ionizing radiation and iodine contrast medium, which may pose a burden for a patient, are the main disadvantages of CEM.

Another diagnostic method used more widely is breast magnetic resonance imaging (BMRI). This method does not require ionizing radiation application but uses magnetic properties of the tissues. Similarly to CEM, focal lesions assessment necessitates administration of the gadolinium contrast medium. Due to the analysis of neoangiogenesis within the tumor in the post-contrast images, the sensitivity of BMRI is very high. Among the high-risk patients, BMRI has much higher sensitivity in comparison to FFDM, but its specificity is lower [22,23,24,25,26,27,28,29]. Despite all the advantages, BMRI’s availability is still low and the costs are high. Screening with BMRI is recommended by the ACR for high-risk women, and for women with extremely dense breasts by EUSOBI (European Society of Breast Imaging) [30,31].

The most common diagnostic method complementary to mammography is HHUS [32,33]. However, this procedure has a few significant limitations: long-lasting process of training individuals performing the examination, the necessity of direct performance by a physician, lack of the possibility of image reproduction, narrow field of view (FOV) and high percentage of false positive results.

Automated breast ultrasonography (ABUS) appeared to be a promising alternative to HHUS due to its multiple advantages. The most important ones are: delegation of the acquisition task to the technicians, possibility of image recording and its multiple reproduction, image processing at the workstations and creating image reconstruction, wider FOV and possibility of computer-aided detection application. Its limitations are: presence of artefacts unique for ABUS, lack of possibility to evaluate vascularization of the focal lesion and its tissue rigidity, lack of possibility to perform biopsy and low availability [34,35,36].

## 2. Materials and Methods

This retrospective analysis comprises 297 women who presented to the Breast Imaging Diagnostics Unit between 2020–2022 in order to undergo a screening mammography examination (FFDM). Additionally, all patients underwent ABUS. If a focal lesion was encountered, CEM was performed. 

The mean age of the patients was 53.6 years, with the youngest patient 29 years old and the oldest 77 years old. ABUS examinations were performed by adequately trained electro-radiology technicians. Each breast was examined in three standard projections: anteroposterior, lateral and medial, and if necessary, additional projections covering upper inner and lower outer quadrants were performed. 

CEM examinations were carried out with the GE Healthcare Senographe Essential. Mammography imaging was performed after 2 min from intravenous contrast medium administration at a dose of 1.5 mL/kg. The examination begun with the healthy breast without any focal lesion detected in the standard bilateral craniocaudal projection (CC), then the second breast was examined; subsequently, mediolateral oblique (MLO) projections were performed in the same order as for CC projections. The apparatus automatically acquired low- and high-energy images, whereas on the diagnostic monitors, low-energy images equivalent to the images obtained with FFDM were available and subtraction images with glandular tissue suppression and visible pathological foci of contrast enhancement. Focal lesions were assessed according to BI-RADS classification.

With FFDM and low-energy CEM, each patient’s breast composition was determined according to ACR classification and the patients with heterogeneously dense or extremely dense breasts (categories C and D) were selected. All patients underwent ABUS and the breast composition type in ABUS was assessed on the basis of ACR BI-RADS as homogenous background echotexture—fat, homogenous background echotexture—fibroglandular and heterogenous background echotexture. In the patients with heterogenous background echotexture, the proportions of fat and fibroglandular tissue were assessed and the group was subdivided into two groups: with the prevalence of fat and with the prevalence of fibroglandular tissue. Then, for the purpose of this study, it was assumed that homogenous background echotexture—fat corresponds with ACR type A, heterogenous background echotexture with the prevalence of fat corresponds with ACR type B, heterogenous background echotexture with the prevalence of fibroglandular tissue corresponds with ACR type C and homogenous background echotexture—fibroglandular corresponds with ACR type D. If a focal lesion was encountered either with FFDM or ABUS, HHUS was also performed. Subsequently, if with FFDM, ABUS and HHUS a suspicious lesion was found, CEM was performed. Each of the examinations were assessed by two experienced radiologists with 5 to 30 years of experience in breast diagnostics.

All focal lesions classified as BI-RADS 4 and 5 were subjected to histopathological verification with the application of core biopsy or vacuum-assisted biopsy and the patients were provided with adequate treatment. The diagnostics and treatment were conducted in one health center.

### 2.1. Histopathological Examinations

Detailed histopathological examinations of the tissue material obtained in biopsy were conducted by a breast cancer dedicated pathologist in the same health center where imaging diagnostics was previously performed. Each sample was fixed in 10% neutral buffered formalin, paraffin-embedded and hematoxylin and eosin stained and finally microscopically evaluated. The lesions were further classified as benign or malignant according to scale B—from 1 to 5. Additional evaluation of estrogen (ER), progesterone (PR), HER2 (*human epidermal growth factor receptor 2*) receptors expression and Ki-67 status, as well as stage and histological type, was performed for malignant lesions. 

### 2.2. Statistical Methods

Statistical analysis of the obtained data was performed using the McNemar’s test for dependent samples, the Z-test for two independent samples and the Chi-square independence test. 

## 3. Results

Breast composition with glandular tissue prevalence with FFDM and low-energy CEM (ACR categories C and D) was determined in 40% of patients (119 patients), including category C in 25% (74 patients) and category D in 15% of patients (45 patients). Figure 3 shows the distribution of patients according to breast composition type according to ACR. 

No focal lesions were detected in 166 patients on both examinations, while in 131 patients, 115 focal lesions were found in one or both examinations. The lesions were histopathologically verified.

With FFDM, 120 focal lesions were identified (73 histopathologically verified), with ABUS, 244 focal lesions (113 histopathologically verified) and with CEM, 103 focal lesions (90 histopathologically verified). Healthy patients were recommended to participate in screening programs at appropriate time intervals according to the guidelines.

Within the group of lesions subjected to verification, invasive cancers were the most numerous—67% (77 lesions). The second in line were DCIS (without invasive cancer component)—10.5% (12 lesions), a group of benign lesions B2—17.5% (20 lesions) and the least numerous group were B3 lesions—of uncertain malignant potential—5% (6 lesions). 

Figure 4 presents the histopathological types of all determined focal lesions.

The patients included in the study were divided into two groups depending on breast composition assessed on the basis of FFDM and low-energy CEM, for the purposes of this work called: group I—composition mostly glandular—ACR C and D category and group II—composition mostly fatty—ACR categories A and B. Diagnostic efficiency of ABUS, FFDM and CEM in relation to breast composition type was compared in both groups based on comparative analysis of the obtained results.

Breast composition types (ACR classification) determined with ABUS appeared to be concordant with FFDM findings in 70%.

The number of focal lesions found with ABUS, FFDM and CEM was assessed in patients with breast category D with multiple focal lesions. The patients with two or more lesions were included in the analysis. Table 1 presents the results.

The correspondence between determining the number of foci with an individual method and histopathology report was also subjected to analysis. This analysis revealed that the highest compliance of imaging with histopathology examination among all methods referred to the patients with breasts with glandular tissue prevalence (categories C and D) with one focus—for FFDM in 82% of cases, for CEM and ABUS in 91% of cases. In patients with two foci confirmed on histopathology, such compliance was present in 33% of cases with FFDM and CEM and 66% with ABUS; in all cases with 3 foci, all lesions were confirmed with FFDM and ABUS but no compliance was found in CEM cases. In the case of one patient with histopathologically proven 4 foci, only CEM visualized all of them. 

The aim of the next stage was to assess which of the diagnostic methods detected more focal lesions in the breast with glandular tissue prevalence (categories C and D). With ABUS, 72% of all lesions were detected, 56% in CEM subtraction images and only 29% in FFDM. ABUS revealed all invasive carcinomas, CEM—83% of them and FFDM—59%. ABUS predominance over CEM and FFDM in all of the above cases was statistically significant. DCIS lesions were visible with ABUS in 100% which is statistically significantly more than with FFDM (59%). However, it was not confirmed that in dense glandular breasts significantly more DCIS lesions were recognized by ABUS than by CEM (93%). ABUS also revealed all LCIS lesions, while CEM revealed 55% and FFDM revealed 63% of them.

Among the patients with breast composition categories C and D, only 3 B3 lesions were found, all visible with ABUS and FFDM and none of them with CEM.

A similar analysis was performed exclusively for the group of patients with extremely glandular breast composition (category D). It appeared that ABUS enabled visualization of 79% of all lesions, CEM revealed 55% and FFDM only 24% of them. Comparable predominance of ABUS over the other two methods in glandular breast type was determined in relation to the particular histopathological results. All invasive carcinomas found in patients with glandular breast composition were visible with ABUS, whereas 83% of them were recognized by CEM and only 44% by FFDM. In the cases of DCIS lesions, 100% were seen with ABUS, 94% with CEM and 44% with FFDM. While ABUS visualized all lesions verified as LCIS, only half of them were visible with FFDM and 33% with CEM. Among the patients with dense glandular breast composition—ACR category D—only one B3 lesion was visible with ABUS and FFDM and none showing contrast enhancement on subtraction images with CEM was found. 

The proportion of all lesions characterized as benign on imaging (BI-RADS 2) was compared between separate methods in the breasts which were mostly fatty (composition categories A and B—group II) and mostly glandular (categories C and D—group I). In group II, ABUS visualized all benign lesions and in group I, 99% of lesions. With FFDM, only 7.6% of benign lesions were found in group II and 4.3% in group I. None of the benign lesions showed post-contrast enhancement with CEM. 

An attempt to establish the presence of significant differences between the characterization of focal lesions diagnosed in patients with breasts qualified to groups I and II was made at the following stage. It brought the conclusion that the size of the cancers determined on the basis of post-operative histopathological examination is larger in the breasts qualified to group I (the size of the majority of the lesions in the breasts qualified as type II ranges between 14 and 26 mm, whereas in breast type I—between 11 and 37 mm). Figure 5 presents the distribution of the focal lesion sizes depending on the breast composition category.

Differences in the incidence of distinguished histological types of focal lesions in breasts mostly fatty or mostly glandular were sought. No correlation between breast composition category and frequency of invasive cancers, DCIS, B3 lesions and benign lesions was found. Similarly, there was no correlation between estrogen and progesterone receptor expression levels, HER2 (human epidermal growth factor receptor 2) and Ki-67.

A separate analysis of sensitivity, specificity, positive and negative predictive values for ABUS, FFDM, CEM and a combination of FFDM and ABUS in both groups was conducted. Table 2 and Table 3 presents the results and Figure 6 and Figure 7 present the ROC curves.

## 4. Discussion

Fibroglandular dense breast type poses an independent risk factor for breast cancer, occurs with high frequency (in the study group, breast categories C and D were present in 40% of patients) and the diagnostic obstacles resulting from that composition type lead to seeking imaging methods which may be alternative or additive to FFDM.

There are no significant differences between sensitivity, specificity, positive and negative predictive values for ABUS, FFDM and CEM among patients with breast types I and II. In group I, the sensitivity, specificity, positive and negative predictive values for FFDM are lower in comparison to group II: FFDM shows much lower sensitivity (85 vs. 94) and specificity (13 vs. 18), PPV (73 vs. 83), NPV (25 vs. 40) and accuracy (66 vs. 80). On the contrary, specificity and NPV of ABUS appeared to be higher in group I than in group II (40 vs. 18 in both cases), while sensitivity, PPV and accuracy only slightly lower (78 vs. 81, 78 vs. 81 and 68 vs. 69, respectively).

The statistics presented below suggest that the breast composition category has a bigger influence on the diagnostic efficiency of FFDM than ABUS, which seems to be relatively independent of this feature and presents more advantageous parameters in some aspects. CEM has the highest accuracy among the evaluated diagnostic methods—higher than ABUS and FFDM in both groups (74 for group I and 85 for group II), yet the specificity of this method is the lowest of all three (8 for group I and 17 for group II vs. 40 and 18 for ABUS and 13 and 18 for FFDM). The combination of ABUS and FFDM results in an improvement of sensitivity compared to the sensitivity of these two methods alone—which for both groups is 100 and is higher than the sensitivity of CEM (100 vs. 93 for both groups). The accuracy of the combination of ABUS and FFDM was the highest of all methods for group I (75 vs. 74 for CEM) and only slightly lower for CEM for group II (81 vs. 85). Low specificity of CEM is associated with the presence of non-specific contrast enhancement foci on subtraction images which may result from benign parenchyma enhancement (BPH) and be related to its intensity. However, breast composition type has a smaller impact on the method’s efficiency than in FFDM. As the above study shows, the diagnostic efficiency of FFDM is highly dependent on breast composition type, which further influences the decrease in sensitivity, specificity, positive and negative predictive values and accuracy in cases of dense fibroglandular breasts. The efficiency of ABUS appeared to be irrespective of breast composition type, which is confirmed by the slight differences of parameters obtained for women both with the prevalence of either fibroglandular or fatty tissue. Among the studied methods, CEM shows high accuracy but relatively low specificity in the group of women with mostly glandular breasts. The accuracy of the combination of ABUS and FFDM is higher than the accuracy of CEM for breasts with the prevalence of fibroglandular tissue and comparable with the accuracy of CEM for breasts with the prevalence of fatty tissue. 

Significantly decreased sensitivity of FFDM in the group of patients with mostly glandular breasts results from obscuring focal lesions by overlapping layers of glandular tissue. The sensitivity of mammography in cases of entirely fatty breasts (ACR category A) is about 98%, while for extremely dense breasts (ACR category D) is only 48% [37]. This may be the reason for detecting so-called interval cancers, even in patients involved in screening programs and undergoing examinations at certain time intervals according to the guidelines—which means cancers diagnosed on a subsequent FFDM performed within the screening scheme. This may indicate the fact that early-stage focal lesions could have been elusive due to breast composition type.

To a large extent, the aforementioned problem is reduced in CEM, where lesions are visualized on subtraction images as post-contrast enhancement foci. The fact of detecting a larger number of foci with ABUS and CEM than FFDM suggests that the above-mentioned methods may complement breast cancer diagnostics in women with dense breasts, particularly in differentiating between unifocal and multifocal breast carcinomas, which is vital for the treatment process and may drastically influence the choice of treatment options. Figure 8a–c show the images of a malignant lesion using different imaging methods on the same patient.

Within the analyzed group of patients with dense breasts (category D), ABUS enabled the detection of the biggest number of foci in multifocal processes (40), exceeding FFDM (13) and even CEM (21). Moreover, ABUS showed the highest compliance in determining the actual number of foci in multifocal processes in relation to histopathology being a golden standard, which emphasizes the role of ABUS as an additive method to FFDM in the course of a proper diagnosis and staging, indispensable for the choice of an adequate treatment. The highest compliance with histopathology in determining the number of focal lesions for all methods referred to the patients with one focus—for CEM and ABUS higher than for FFDM.

Data obtained in the study demonstrate a significant prevalence of ABUS in imaging foci of various types—various malignancy potential—in the breasts which are mostly glandular (categories C and D). ABUS revealed all invasive cancers, DCIS, LCIS and B3 lesions. On the contrary, a strikingly low detection rate was shown for FFDM in relation to breast category D, not involving even half of invasive cancers for DCIS and only a half for LCIS. That fact proves the inefficiency of FFDM within this group of patients and suggests the necessity to complement it with other and more accurate methods. The lesion detection rate with CEM subtraction images appeared to be higher than FFDM in the majority of cases, similar to ABUS. The most obvious prevalence of ABUS over the other methods was visible in cases of benign lesions, regardless of breast composition type, yet were clinically less significant. The analysis presented above shows that ABUS, without the need for ionizing radiation or contrast medium application and not posing a burden for a patient, is a method of high potential to be used as an additive to FFDM in dense breast evaluation.

It was confirmed that the size of cancers determined on the post-operative histopathology examination is larger in breasts with fibroglandular tissue prevalence than in mostly fatty breasts, which suggests that lesions detected in breasts of this type were more advanced-stage cancers. This correlation may result from the fact that lesions present in dense breasts become clinically symptomatic later in comparison to lesions found in patients with other types of breasts. Furthermore, it results in the hindered detection of such lesions on palpable self-examination and lower sensitivity of widely available diagnostic methods (particularly FFDM) among patients with dense breasts.

Numerous studies considering the role of handheld ultrasound in breast cancer detection in women with dense breasts were conducted, revealing its importance as a complementary method. Mengmeng et al. compared the diagnostic efficiency of ABUS and handheld ultrasound and stated that both methods allowed for an increase in breast cancer detection in women with dense breasts. Taking into consideration multiple advantages of ABUS in comparison to handheld ultrasound, such as independence from an operator and reproduction, we may indicate a large potential of ABUS in breast cancer detection within that group of patients [38].

The added value of ABUS in comparison with handheld ultrasound is the possibility of AI application, which is proven to bring good results, helps to save the radiologist’s time, may prevent overlooking malignant lesions, and enables increasing the radiologist’s specificity without decreasing sensitivity [39]. As it was shown in the studies, the evaluation of ABUS images is characterized by high inter-observer reliability, even despite the lack of clinical data or other imaging data [40].

There are numerous reports in the literature referring to the increase in FFDM diagnostic efficiency due to the combination with ABUS. Wilczek et al. proved that adding FFDM to ABUS enabled the detection of 1.8 more cancers in comparison with FFDM alone in patients with dense breasts [41]. In the Somoinsight study by Brem et al., in a large group of 15318 women, they showed that the combination of FFDM and ABUS allowed the detection of 1.9 more cancers per 1000 patients with dense breasts [42]. In the study performed by Kelly et al., the ABUS and FFDM combination made it possible to double cancer detection in women with dense breasts [43]. In a subsequent study, Giuliano et al. stated that the addition of ABUS to FFDM facilitated the detection of 7.7 more cancers per 1000 patients with dense breasts [44]. Then Giger et al. proved that the combined use of ABUS and FFDM allowed to obtain an AUC of 0.82 in comparison to an AUC of 0.72 for FFDM only [45].

Determining the breast composition category with ABUS appeared to be consistent with the ACR evaluation on the basis of FFDM in 70%, which suggests that ABUS as an ultrasound method is an adequate tool for determining breast composition type. This means that classifying the patient as having dense breasts and a higher risk of developing breast cancer may be performed without the necessity to undergo FFDM routinely after 50 years of life, and as a result, adequately distinguishes high-risk patients from the evaluated group. A few reports on the potential application of ABUS in determining breast density have already appeared. Woo et al. compared determining breast density and volume in ABUS and MRI, obtaining high consistency [46]. In another study, also by Woo et al., the authors used rapid breast volume and density analysis in ABUS, obtaining satisfactory effects and describing it as an effective tool for breast density assessment [47].

ABUS is a breast imaging modality that is not likely to replace other breast cancer screening methods, such as FFDM. Nevertheless, numerous scientific reports suggest that it may be a valuable complementary tool to FFDM, particularly for the diagnosis of cancer in women with dense breasts. Our study, however, supports prior reports that ABUS presents several advantages, e.g., non-invasiveness, short performance time, reproducibility and increased diagnostic accuracy in combination with FFDM, that make it a valuable complementary screening tool, particularly for women with dense breasts. 

## Figures and Tables

**Figure 1 biomedicines-11-03226-f001:**
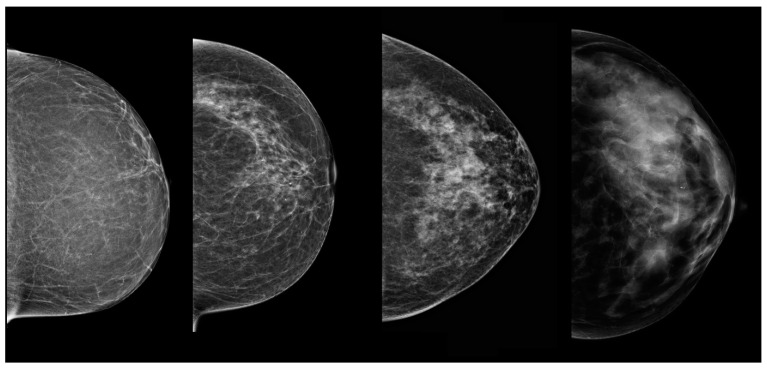
Breast composition on craniocaudal projections with mammography and low-energy CEM: types A, B, C and D (from the left).

**Figure 2 biomedicines-11-03226-f002:**
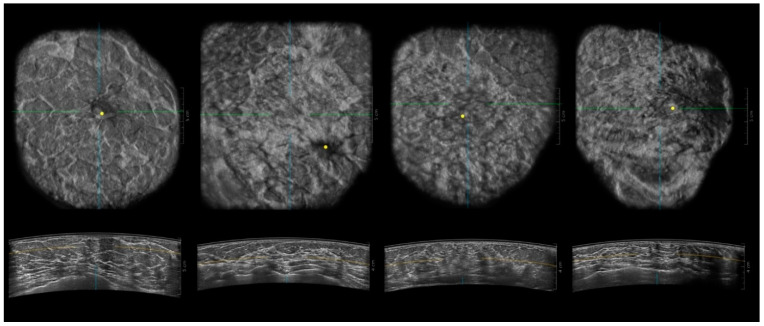
ABUS coronal and transverse images of breasts according to ACR breast composition type with FFDM: A, B, C and D (from the left). The upper line represents coronal images and the lower transverse images. Lines show cursor position and yellow dot is the nipple.

**Figure 3 biomedicines-11-03226-f003:**
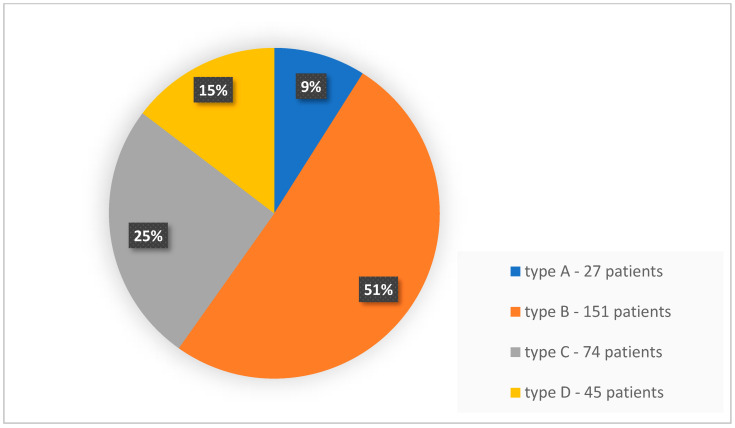
Distribution of breast composition types with FFDM and low-energy CEM according to ACR within the examined group.

**Figure 4 biomedicines-11-03226-f004:**
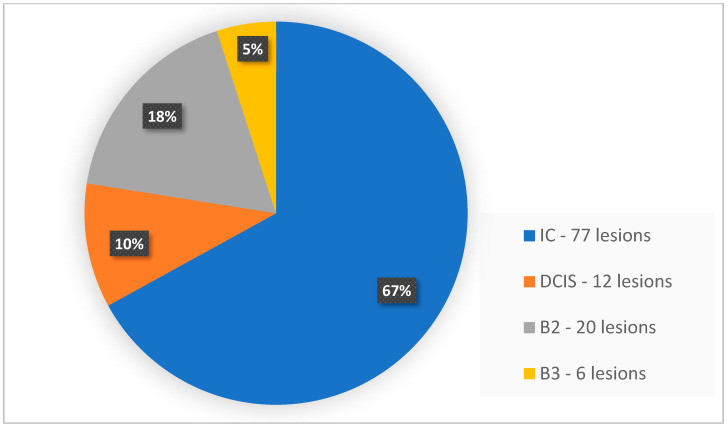
Histopathological types of lesions determined in patients included in the study.

**Figure 5 biomedicines-11-03226-f005:**
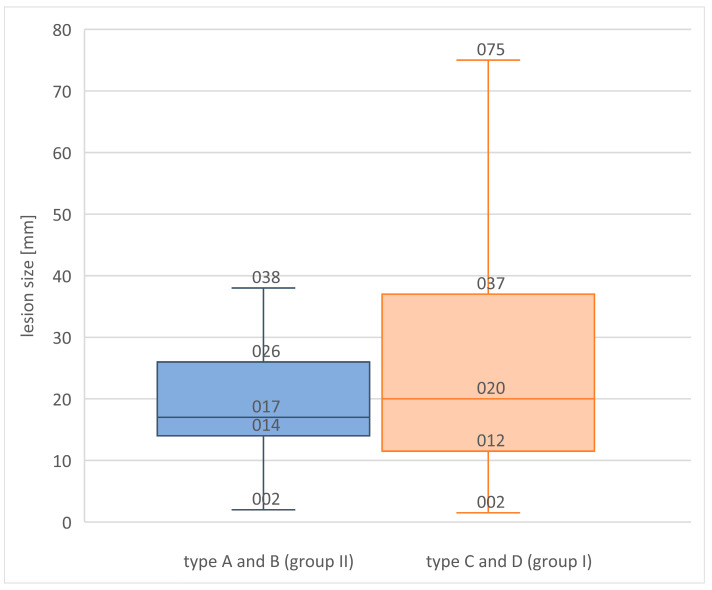
Size of the focal lesion on histopathology in relation to breast composition category.

**Figure 6 biomedicines-11-03226-f006:**
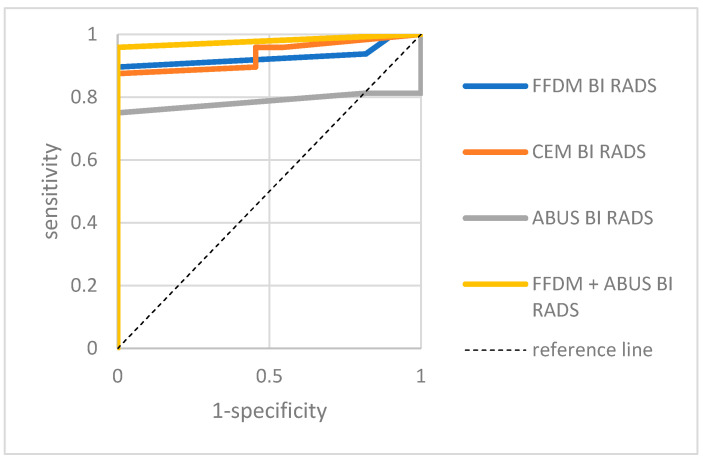
ABUS, FFDM, CEM and combined ABUS and FFDM ROC curves in patients with breast ACR categories C and D—group I. ABUS: AUC = 0.805, *p* = 0.0003; FFDM: AUC = 0.793, *p* = 0.0004, CEM: AUC = 0.877, *p* = 0.0000; ABUS and FFDM: AUC = 0.921, *p* = 0.0000.

**Figure 7 biomedicines-11-03226-f007:**
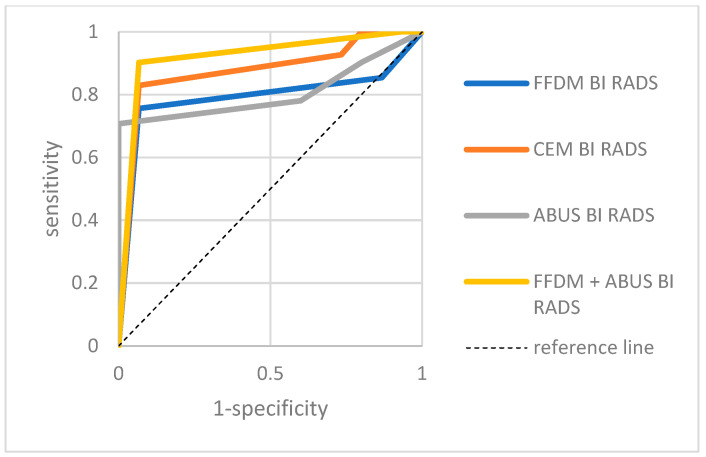
ABUS, FFDM, CEM and combined ABUS and FFDM ROC curves in patients with breast ACR categories A and B—group II. ABUS: AUC = 0.787, *p* = 0.0016; FFDM: AUC = 0.929, *p* = 0.0000, CEM: AUC = 0.935, *p* = 0.0000; ABUS and FFDM: AUC = 0.979, *p* = 0.0000.

**Figure 8 biomedicines-11-03226-f008:**
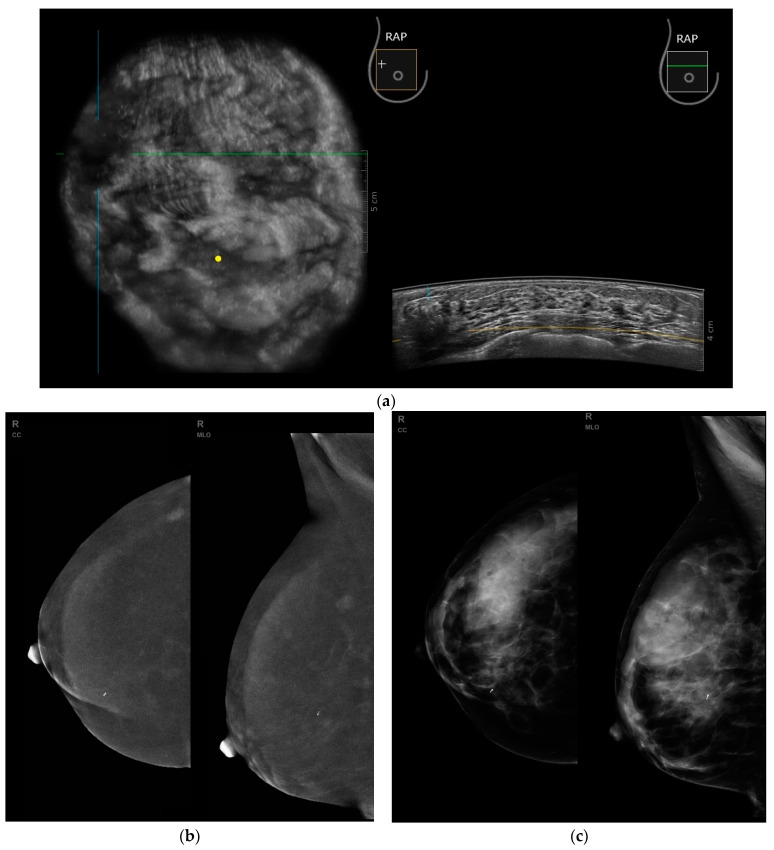
(**a**–**c**) ABUS and CEM images show verified right breast cancer. (**a**) Coronal and transverse planes, visible hypoechoic mass between the lines. (**b**) Subtraction CEM image in RCC and RMLO projections, visible oval enhancing mass in the upper outer quadrant. (**c**) FFDM image in RCC and RMLO projections, mass not visible.

**Table 1 biomedicines-11-03226-t001:** Distribution of focal lesions found with FFDM, ABUS and CEM in patients with ACR breast category D.

No. of Lesions in a Patient (Any Method)	No. of Patients	Total No. of Lesions	Lesions Visible with FFDM	Lesions Visible with ABUS	Lesions Visible with CEM
2	3	6	2 cases (2 patients with 1 lesion)	5 cases (including 2 patients with 2 lesions and 1 patient with 1 lesion => 3 patients)	5 cases (including 1 patient with 2 lesions and 3 patients with 1 lesion => 4 patients)
3	4	12	4 cases (including 2 patients with 2 lesions)	10 cases (3 patients with 3 lesions and 1 patient with 1 lesion => 4 patients)	4 cases (including 1 patient with 2 lesions and 2 patients with 1 lesion => 3 patients)
4	5	20	6 cases (including 2 patients with 2 lesions and 2 patients with 1 lesion => 4 patients)	15 cases (including 2 patients with 4 lesions, 1 patient with 3 lesions and 2 patients with 2 lesions => 5 patients)	11 cases (including 1 patient with 4 lesions, 2 patients with 3 lesions and 1 patient with 1 lesion => 4 patients)
5	2	10	1 case (1 patient with 1 lesion)	10 cases (2 patients with 5 lesions)	1 case (1 patient with 1 lesion)
Total	14	48	13 cases => 9 patients	40 cases => 14 patients	21 cases (12 patients)

**Table 2 biomedicines-11-03226-t002:** Comparison of sensitivity, specificity, positive and negative predictive values for ABUS, FFDM and CEM in patients with breast ACR categories C and D—group I.

Specification	Sensitivity	Specificity	PPV	NPV	Accuracy
ABUS	78.05 [65.38; 90.72]	40 [15.21; 64.79]	78.05 [65.38; 90.72]	40 [15.21; 64.79]	67.86 [55.63; 80.09]
FFDM	85.37 [74.55; 96.18]	13.33 [0; 30.54]	72.92 [60.34; 85.49]	25 [0; 55.01]	66.07 [53.67; 78.47]
CEM	92.68 [84.71; 100]	8.33 [0; 23.97]	77.55 [65.87; 89.23]	25 [0; 67.44]	73.58 [61.72; 85.45]
FFDM + ABUS	100	6.67 [0; 19.29]	74.55 [63.03; 86.06]	100	75 [0; 100]
*p* (ABUS vs. FFDM)	0.606	0.22	0.575	0.472	0.841
*p* (ABUS vs. CEM)	0.149	0.37	0.954	0.567	0.512
*p* (FFDM vs. CEM)	0.248	1	0.597	1	0.393
*p* (FFDM + ABUS vs. CEM)	0.248	0.479	0.719	0.171	0.866
*p* (FFDM + ABUS vs. FFDM)	0.04	1	0.851	0.134	0.299

**Table 3 biomedicines-11-03226-t003:** Comparison of sensitivity, specificity, positive and negative predictive values for ABUS, FFDM and CEM in patients with breast ACR categories A and B—group II.

Specification	Sensitivity	Specificity	PPV	NPV	Accuracy
ABUS	81.25 [70.21; 92.29]	18.18 [0; 40.97]	81.25 [70.21; 92.29]	18.18 [0; 40.97]	69.49 [57.74; 81.24]
FFDM	93.75 [86.9; 100]	18.18 [0; 40.97]	83.33 [73.39; 93.27]	40 [0; 82.94]	79.66 [69.39; 89.93]
CEM	93.48 [86.34; 100]	16.67 [0; 46.49]	89.58 [80.94; 98.23]	25 [0; 67.44]	84.62 [74.81; 94.42]
FFDM + ABUS	100 [100; 100]	-	81.36 [71.42; 91.29]	-	81.36 [71.42; 91.29]
*p* (ABUS vs. FFDM)	0.150	0.617	0.783	0.350	0.205
*p* (ABUS vs. CEM)	0.150	0.480	0.247	0.770	0.056
*p* (FFDM vs. CEM)	1	1	0.359	0.635	0.498
*p* (FFDM + ABUS vs. CEM)	0.248	-	0.235	-	0.816
*p* (FFDM + ABUS vs. FFDM)	0.248	-	0.783	-	0.649

## Data Availability

Unavailable due to privacy.

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
