# Peer review of "Comparative Analysis of Diagnostic Performance of Automatic Breast Ultrasound, Full-Field Digital Mammography and Contrast-Enhanced Mammography in Relation to Breast Composition"

_biomedicines, 2023, doi:10.3390/biomedicines11123226_

Round 1

Reviewer 1 Report

Comments and Suggestions for Authors

This paper compares the diagnostic performance of three breast imaging modalities, FFDM, CEM, and ABUS, as a function of breast composition.

The goal is of interest because, as the authors also point out in the introduction, breast composition presents challenges in breast cancer detection. Dense breasts often require complementary imaging as the standard FFDM is not sufficient.

Unfortunately, the organization and presentation of the data is extremely poor and does not justify any conclusions. Basic elements of the study such as the description of the database, the classification process, the distribution of lesions, the breast composition per modality are not clear and as presented are incomprehensible. 

The ROC curves are not useful at all as presented (why aren't any ROC metrics used, such as the AUC?), Tables 2 and 3 show no differences although it is not clear what is compared, and no conclusion may be derived from any of these.

Another major weakness is the description of the breast composition. It is understood that the BIRADS classification was applied to FFDB and to the low energy CEM (this was assumed as there is no indication in the paper), different number of patients for each one. Should there be a description of the agreement/disagreement between the two for the common number of patients?  In addition, according to the US lexicon, the characterization of breast composition in the ABUS images is done in three categories: homogeneous-fat, homogeneous-fibroglandular, c-heterogeneous. How is this comparable to the FFDM and CEM? How did the authors distinguish four categories instead of the recommended three? What criterion did they follow?

Overall, the study needs major re-organization, clearer presentation, focused discussion, and realistic conclusions.  

Comments on the Quality of English Language

Acceptable, minor issues in grammar and syntax.

Author Response

Dear Reviewer,

Thank you for your insightful comments!

Regarding concerns of Reviewer 1:

This paper compares the diagnostic performance of three breast imaging modalities, FFDM, CEM, and ABUS, as a function of breast composition.

The goal is of interest because, as the authors also point out in the introduction, breast composition presents challenges in breast cancer detection. Dense breasts often require complementary imaging as the standard FFDM is not sufficient.

Unfortunately, the organization and presentation of the data is extremely poor and does not justify any conclusions. Basic elements of the study such as the description of the database, the classification process, the distribution of lesions, the breast composition per modality are not clear and as presented are incomprehensible. 

The ROC curves are not useful at all as presented (why aren't any ROC metrics used, such as the AUC?), Tables 2 and 3 show no differences although it is not clear what is compared, and no conclusion may be derived from any of these.

The AUC values were added.

Another major weakness is the description of the breast composition. It is understood that the BIRADS classification was applied to FFDB and to the low energy CEM (this was assumed as there is no indication in the paper), different number of patients for each one. Should there be a description of the agreement/disagreement between the two for the common number of patients?  In addition, according to the US lexicon, the characterization of breast composition in the ABUS images is done in three categories: homogeneous-fat, homogeneous-fibroglandular, c-heterogeneous. How is this comparable to the FFDM and CEM? How did the authors distinguish four categories instead of the recommended three? What criterion did they follow?

All the patients in the study underwent FFDM and ABUS, patients with focal lesions underwent also CEM. The ACR breast type was assessed on the basis on FFDM and low energy CEM. The major analysis regarding the breast type was conducted on the basis of FFDM and low energy CEM breast composition type: A, B, C, D. We analyzed the diagnostic performance of FFDM, CEM and ABUS in breasts with different types of composition in FFDM. On ABUS we assessed the breast composition according to US lexicon - homogeneous-fat, homogeneous-fibroglandular, heterogeneous, but we didn’t analyzed the data according to these subdivisions. In the patients with heterogenous background echotexture on ABUS the proportions of fat and fibroglandular tissue was assessed and the group was subdivided in to two groups; with the prevalence of fat and with the prevalence of fibroglandular tissue. Then, for the purpose of this study, it was assumed that homogenous background echotexture - fat corresponds with ACR type A, heterogenous background echotexture with the prevalence of fat corresponds with ACR type B, heterogenous background echotexture with the prevalence of fibroglandular tissue corresponds with ACR type C and homogenous background echotexture - fibroglandular corresponds with ACR type D.

Overall, the study needs major re-organization, clearer presentation, focused discussion, and realistic conclusions.  

Best regards,

Authors

Reviewer 2 Report

Comments and Suggestions for Authors

Comparative Analysis of Diagnostic Performance of Automatic Breast Ultrasound, Full Field Digital Mammography and Contrast Enhanced Mammography in Relation to Breast Composition

Comments: 

Abstract

    • 14: Indicate the meaning of ACR

Introduction

    • 34: Include the reference 2 within the paragraph.

    • Figure 1: Include in the description of the image the projection of the mammograms.

    • 39: Remove meaning of ACR acronyms, previously it should have been included in the abstract because the acronyms are mentioned there.

    • 82: Add reference to The European Commission Initiative on Breast Cancer (ECIBC) and indicate the meaning of the acronym.

    • 106-107: Remove the meaning of HHUS and just leave the acronym, previously mentioned in line 76.

Materials and methods

    • No comments

Results

    • 163-165: As a suggestion, add in the text the number of patients according to their percentage of sample distribution according to the ACR.

    • Figure 3: As a suggestion, include the number of patients according to ACR distribution in the pie chart.

    • 176-180: As a suggestion, add in the text the number of patients according to the percentage distribution of the diagnosis that the patients received.

    • Figure 4: As a suggestion, include the number of patients according to the percentage distribution of the diagnosis that the patients received.

    • Figure 5: Include on the Y-axis (lesion size) in parentheses mm (mm), also include on the X-axis the groups (group I and II), in addition to the types.

    • 248-249: The ROC curves are represented in Figures 6 and 7. Correct in text and figure captions.

    • 316: Should be figures 8 (8a, 8b and 8c). Correct in text and figure captions.

    • Figure 8 (already corrected): As a suggestion, find a way in which the figures are not on different pages and put the reference of "a, b and c" in the corresponding image.

Discusión

    • No comments

Comments on the Quality of English Language

Minor spelling issues

Author Response

Dear Reviewer,

Thank you for your insightful comments!

Regarding concerns of Reviewer 2:

Abstract

  • 14: Indicate the meaning of ACR

The meaning of ACR was added.

Introduction

  • 34: Include the reference 2 within the paragraph.

The reference was added within the paragraph.

  • Figure 1: Include in the description of the image the projection of the mammograms.

The description of the projection of the mammograms was added.

  • 39: Remove meaning of ACR acronyms, previously it should have been included in the abstract because the acronyms are mentioned there.

The meaning of ACR acronyms was removed.

  • 82: Add reference to The European Commission Initiative on Breast Cancer (ECIBC) and indicate the meaning of the acronym.

The reference was added.

  • 106-107: Remove the meaning of HHUS and just leave the acronym, previously mentioned in line 76.

The meaning of HHUS was removed.  

Materials and methods

  • No comments

Results

  • 163-165: As a suggestion, add in the text the number of patients according to their percentage of sample distribution according to the ACR.

The numbers of patients were added.

  • Figure 3: As a suggestion, include the number of patients according to ACR distribution in the pie chart.

The numbers were added.

  • 176-180: As a suggestion, add in the text the number of patients according to the percentage distribution of the diagnosis that the patients received.

The numbers were added.

  • Figure 4: As a suggestion, include the number of patients according to the percentage distribution of the diagnosis that the patients received.

The numbers were added.

  • Figure 5: Include on the Y-axis (lesion size) in parentheses mm (mm), also include on the X-axis the groups (group I and II), in addition to the types.

The description was reviewed and updated.

  • 248-249: The ROC curves are represented in Figures 6 and 7. Correct in text and figure captions.

The text was reviewed and corrected.

  • 316: Should be figures 8 (8a, 8b and 8c). Correct in text and figure captions.
  • Figure 8 (already corrected): As a suggestion, find a way in which the figures are not on different pages and put the reference of "a, b and c" in the corresponding image.

The text and figures captions and the presentation of the figures were reviewed and corrected.

We hope the corrections will make the article better and more valuable to the readers.

Best regards,

Authors

Round 2

Reviewer 1 Report

Comments and Suggestions for Authors

line 13 - Use "American College of Radiology (ACR)" and not the other way around, the same way you list all other abbreviations.

line 278 - use "in Figure" instead of "in the Figure".

Decide ... is it going to be P or p for the propability? See Tables and Figure 6.

lines 426-428 - instead a "pose" suggest simply "be" and generally this is not a good sentence in english ... complentary to what and the diagnostic gap of FFDM ... where is that gap? Just simplify and avoid colloquialisms. For example: "... it may be a valuable complementary tool to FFDM particularly for the diagnosis of cancer in women with dense breasts."

line 429 - suggest "reproducibility" instead of "repeatability"

lines 425-430 - this is your punchline and final conclusion and should include findings of your study as well.  Avoid strong statements that include "should" and again be clearer. For example:

"ABUS is a breast imaging modality that is not likely to replace other breast cancer screening methods, such as FFDM. Our study, however, supports prior reports that ABUS presents several advantages, e.g., non-invasiveness, short performance time, reproducibility, increased diagnostic accuracy in combination with FFDM, that make it a valuable complementary screening tool particularly for women with dense breasts. 

Comments on the Quality of English Language

See Suggestions to Authors.

Author Response

Dear Reviewer,

Thank you for your insightful comments!

Regarding concerns of Reviewer 1:

Line 13 - Use "American College of Radiology (ACR)" and not the other way around, the same way you list all other abbreviations.

We have revised the text.

line 278 - use "in Figure" instead of "in the Figure".

We have revised the text as well as grammar.

Decide ... is it going to be P or p for the propability? See Tables and Figure 6.

We have revised the text.

lines 426-428 - instead a "pose" suggest simply "be" and generally this is not a good sentence in english ... complentary to what and the diagnostic gap of FFDM ... where is that gap? Just simplify and avoid colloquialisms. For example: "... it may be a valuable complementary tool to FFDM particularly for the diagnosis of cancer in women with dense breasts."

We have revised the text as well as grammar.

line 429 - suggest "reproducibility" instead of "repeatability"

We have revised the text.

lines 425-430 - this is your punchline and final conclusion and should include findings of your study as well.  Avoid strong statements that include "should" and again be clearer. For example:

"ABUS is a breast imaging modality that is not likely to replace other breast cancer screening methods, such as FFDM. Our study, however, supports prior reports that ABUS presents several advantages, e.g., non-invasiveness, short performance time, reproducibility, increased diagnostic accuracy in combination with FFDM, that make it a valuable complementary screening tool particularly for women with dense breasts. 

We have revised the text.

We hope the corrections will make the article better and more valuable to the readers.

Best regards,

Authors

Reviewer 2 Report

Comments and Suggestions for Authors

The authors have tackled and resolved all the issues that were raised in the previous revision, demonstrating their dedication to improving the quality of the manuscript.

Comments on the Quality of English Language

Minor spelling issues

Author Response

Dear Reviewer,

Thank you for your insight, we corrected all spelling.

Best regards,

Authors